# Effect of the Cu^2+/1+^ Redox Potential of Non-Macrocyclic Cu Complexes on Electrochemical CO_2_ Reduction

**DOI:** 10.3390/molecules28135179

**Published:** 2023-07-03

**Authors:** Kyuman Kim, Pawel Wagner, Klaudia Wagner, Attila J. Mozer

**Affiliations:** Intelligent Polymer Research Institute and ARC Centre of Excellence for Electromaterials Science, University of Wollongong, Wollongong, NSW 2522, Australia; kk816@uowmail.edu.au (K.K.); pawel@uow.edu.au (P.W.); kwagner@uow.edu.au (K.W.)

**Keywords:** Cu complexes, redox potential, electrochemical CO_2_ reduction

## Abstract

Cu^2+/1+^ complexes facilitate the reduction of CO_2_ to valuable chemicals. The catalytic conversion likely involves the binding of CO_2_ and/or reduction intermediates to Cu^2+/1+^, which in turn could be influenced by the electron density on the Cu^2+/1+^ ion. Herein we investigated whether modulating the redox potential of Cu^2+/1+^ complexes by changing their ligand structures influenced their CO_2_ reduction performance significantly. We synthesised new heteroleptic Cu^2/1+^ complexes, and for the first time, studied a (Cu-bis(8-quinolinolato) complex, covering a Cu^2+/1+^ redox potential range of 1.3 V. We have found that the redox potential influenced the Faradaic efficiency of CO_2_ reduction to CO. However, no correlation between the redox potential and the Faradaic efficiency for methane was found. The lack of correlation could be attributed to the presence of a Cu-complex-derived catalyst deposited on the electrodes leading to a heterogeneous catalytic mechanism, which is controlled by the structure of the in situ deposited catalyst and not the redox potential of the pre-cursor Cu^2+/1+^ complexes.

## 1. Introduction

The rational design of metal complexes for CO_2_ reduction relies on developing a clear understanding of the correlations between their molecular structure and catalytic performance [1]. The structure of the complexes can be altered in several ways, including changing the coordination number of the metal centre [2,3], the number of ligand binding sites (denticity) [4,5], and the electronic charges on the ligands [6,7]. Furthermore, the strength of the coordination bond, i.e., the coupling of the frontier orbitals of the metal centre and the ligand binding site, can be altered by placing electron-donating or -withdrawing groups on the ligands [8,9]. Steric groups on the ligands can affect the length of the coordination bond, which modifies the electronic coupling between the ligand and the metal ion. A prime example is the approximately 0.3 V difference in the redox potential of structurally similar ortho- and para-methyl-substituted tetragonal Cu-bipyridine complexes [10]. Due to the steric hinderance of the methyl groups at the ortho-position, the Cu^2+/1+^-N bond is weaken, and the electron density on the Cu^2+/1+^ is lowered as compared to that of the para-substituted bipyridyl ligand. Such Cu^2+/1+^ complexes have been extensively used in both homogeneous and heterogeneous electrocatalytic CO_2_ reduction experiments [1]. Cu complexes are one of the rare metal complexes that can reduce CO_2_ to methane (CH_4_) as well as to C-C products of ethylene (C_2_H_4_) and ethanol (C_2_H_5_OH) with high selectivity. CO_2_ reduction experiments using Cu complexes suggested that the catalytic mechanism of CO_2_ reduction involved the binding of the CO_2_ or its intermediates to the Cu^2+/1+^ metal centre [11,12,13]. However, to date, no clear correlations between the redox potential of tetragonal Cu complexes and their CO_2_ reduction performance have been established [1].

On the other hand, correlations between the molecular structure of non-Cu-based complexes and their CO_2_ reduction performance have been studied in homogeneous (complexes dissolved in organic electrolyte) catalysis [9,14,15,16]. The influence of electron-donating/withdrawing groups placed on the ligands of porphyrin and bipyridine in CO_2_ reduction has been studied [8,14]. These reports suggested that electron-withdrawing groups (positive shift of the redox potential of the metal ion) led to a positive shift of the onset potential for CO_2_ reduction [8,14]. However, the turnover frequency (TOF) of the catalytic conversion to CO decreased due to the less nucleophilic character of the metal centre induced by the electron-withdrawing group. On the contrary, electron-donating groups enhanced the nucleophilicity of the metal centre and increased the binding ability of CO_2_, but a negative shift of the onset potential was found [14].

Cu^2+/1+^ complexes can perform CO_2_ reduction by both homogeneous and heterogeneous catalytic mechanisms [1]. Homogeneous catalysis is defined here as involving the dissolved Cu^2+/1+^ complexes as active sites for catalysis. These molecular catalysts would be transported from the bulk of the electrolyte to the working electrode by diffusion. Heterogenous catalysis is considered if the catalytic active site is located in a solid catalyst layer. In this case the Cu^2+/1+^ complex acts as precursor to reversibly [17,18,19] or irreversibly [20,21,22] form metal particles and/or an organic phase containing the ligands or ligan fragments. Switching the catalytic mechanism to heterogeneous during CO_2_ reduction performance testing in nominally homogeneous catalysis can significantly alter product selectivity and Faradaic efficiency (FE). Therefore, the tendency of forming such heterogeneous catalyst layers has to date prevented studies of the correlation between the redox potential of Cu^2+/1+^ complexes and their CO_2_ reduction performance in homogeneous catalysis [1]. With regard to the heterogeneous metal catalyst phase, the relationship between the product selectivity of CO_2_ reduction and the binding energy of adsorbed CO (*CO) has been well established [23,24,25,26]. Metals with strong binding of *CO, such as Pt, Ni and Fe, show mainly hydrogen evolution due to “poisoning” of the active site by *CO. Metals with too-weak binding of *CO, such as Au, Ag and Zn, produce CO as the main CO_2_ reduction product, due to the early release of CO before hydrogenation can occur. Cu metal has an intermediate binding strength of *CO, which can facilitate further reduction steps of *CO to hydrocarbons. In addition to the metallic phase, nitrogen atoms of organic molecules are known to bind or “fix” CO_2_ [27]. Coordinating ligands to Cu metal can further enhance the hydrogeneration of *CO to CH_4_ before releasing CO [28]. Therefore, the chemical nature of the organic content, the coordination number of the N-containing phase of the Cu-complex-derived catalyst in particular, may also have an effect on the product selectivity and CO_2_ reduction performance. The electrochemical stability of the Cu complexes at the highly reducing negative potentials during CO_2_ reduction experiments is expected to be influenced by the redox potential of the complexes. In turn, the deposition and the chemical nature/structure of the Cu-complex-derived catalyst influencing CO_2_ reduction performance may be influenced by the redox potential of the complex.

This work has two aims, as illustrated in Figure 1a–c. By designing and synthesizing Cu^2+/1+^ complexes (Figure 1a) with widely varying redox potential, we aim to determine whether the redox potential of the complexes has any significant effect on CO_2_ reduction performance when the complexes are dissolved in the electrolyte (Figure 1b). At this step, both homogeneous and heterogeneous catalytic mechanisms are possible. We aim to distinguish between the two mechanisms by finding correlations between the redox potential of the Cu^2+/1+^ complexes and the ratio of the CO_2_ reduction products initially (first 20 min) and at the end (80 min) of CO_2_ reduction experiments in the presence of Cu^2+/1+^ complexes. Then, in the next step (Figure 1c), the working electrodes with the deposited catalyst layers are transferred to a fresh electrolyte with no added Cu^2+/1+^ complexes in the electrolyte (purely heterogeneous mechanism). Then, we again correlate the product ratio during CO_2_ testing with the Cu^2+/1+^ redox potential as well as the product ratios in the homogeneous/heterogeneous mixed mechanism testing.

To achieve the goals, five Cu complexes were designed and synthesised (dimethylphenanthroline Cu complex (*dmp*), Cu complexes of the 6,6′-dimesityl-2,2′-bipyridine ligand and a series of second ligands, phenanthroline (*hp*), bipyridine (*hbpy*), and terpyridine (*htpy*) and Cu-bis(8-quinolinolato) (*cq*)), as shown in Figure 1a. From this study we have excluded macrocyclic Cu complexes such as porphyrins as their redox processes in the negative regime is typically dominated by the ligand reduction and not the Cu^2+/1+^ centre. The homoleptic Cu complex (dimethylphenanthroline Cu complex (*dmp*)) is well-studied in the literature [29]. We have recently shown highly selective CO_2_ reduction to methane in an organic electrolyte in the presence of organic cations TBAP [28]. In DMF solvent, a Cu-complex-derived catalyst formed on the surface of the electrode during chronoamperometric testing under CO_2_ atmosphere. The optimum applied voltage was −2.17 V versus Fc/Fc^+^, where high FE for methane production was observed without significant decomposition of the organic electrolyte [28]. XRD measurements showed the presence of a crystalline Cu phase, while XPS measurements suggested the presence of a N-containing organic phase. We noted that the FE of methane production using the *dmp* complex varied from 80% to 40%, even though the deposition was performed under nominally identical conditions. In addition, we have studied the in situ growth of the Cu-complex-derived catalyst during chronoamperometric testing under a CO_2_ atmosphere [30]. Evidenced by a gradually increasing capacitance, the appearance and growth of new redox active peaks in the CVs, as well as switching the main CO_2_ reduction product from CO (first 5 to 20 min) to CH_4_ (>40 min), the CO_2_ reduction mechanism was explained by switching from homogeneous to a heterogeneous mechanism. Some differences in the morphology of the deposited catalyst were observed, but the origin of the morphology change was not known. For the purposes of establishing correlations between the redox potential and the CO_2_ reduction performance, we included data sets for both high FE (*dmp-A*) and lower FE (*dmp-B*) samples. In addition, we have synthesised three new heteroleptic Cu complexes using the HETPHEN strategy. The HETPHEN strategy (HETeroleptic PHENanthroline) was developed to modulate the electronic nature of tetragonal Cu complexes [31,32] by employing bulky ligands, i.e., by adding phenyl rings at the 2- and 9-position to the phenanthroline or bipyridine [33,34]. The HETPHEN ligands hinder the formation the homoleptic HETPHEN Cu complex, leading to the isolated heteroleptic Cu complex. This strategy has been used to fully characterize supramolecular Cu complexes [35,36]. The key benefit of the HETPHEN strategy for this work is the ability to tune the redox potential of Cu^2+/1+^ using the isolated heteroleptic Cu complexes. The addition of a second, less bulky ligand, can potentially create a lower steric effect at a certain position of the Cu site to facilitate the bonding of CO_2_, leading to selective CO_2_ reduction. The steric effect generated by heteroleptic Cu complexes would also influence the chemical nature of the Cu-complex-derived catalyst during electrolysis, such as the Cu-Cu coordination number, related to the hydrogeneration of *CO to CH_4_. HETPHEN Cu complexes are therefore used for the first time for electrochemical CO_2_ reduction. We used a 6,6′-dimesityl-2,2′-bipyridine ligand and as the second ligand, and we chose phenanthroline (*hp*), bipyridine (*hbpy*), and terpyridine (*htpy*). Unlike bipyridine, terpyridine is tridentate, so the third N atom of the ligand may block the 5th coordination site of the Cu^2+^ complexes. This 5th coordination site may otherwise be occupied by solvent molecules or coordinating counter ions such as Cl^−^ [29], affecting the redox potential significantly. The counter ion for the complexes were bis(trifluoromethane)sulfonimide (TFSI) for *hp* and tetrafluoroborate (BF_4_) for *hbpy* and *htpy*. The synthesis and characterization of new Cu complexes can be found in Section 3, and UV-vis spectra is provided in Appendix A. Lastly, a (Cu-bis(8-quinolinolato)) (*cq*) complex was also tested. In this complex, the Cu^2+^ ion is coordinated with two N atoms and two O atoms. Unlike the polypyridyl complexes, the (Cu^2+^-bis(8-quinolinolato)) complex is overall neutral as the ligand is negatively charged after deprotonation. Its redox potential has been reported to be more negative than the Cu bipyridyl complexes although their geometry is similar. Pape et al. reported that the Cu^2+^/Cu^1+^ redox potential in (Cu-bis(8-quinolinolato) has shifted significantly to −0.564 V vs. NHE (−1.24 V vs. Fc/Fc^+^), a more than 1 V shift as compared to that of the Cu^2+^ salt, measured in 0.01 M TBAClO_4_/DMSO [37].

First, cyclic voltammetry (CV) was performed to consistently determine the redox potentials of the five Cu complexes dissolved in 0.1 M TBAP/DMF electrolyte with 0.1 M water in an argon atmosphere. Then, chronoamperometry (CA) was performed at −2.17 V vs. Fc/Fc^+^ under a CO_2_ atmosphere while the gas products were detected using gas chromatography (GC) at regular time intervals. After chronoamperometric measurements, the carbon paper electrodes with Cu-complex-derived catalyst layers deposited were rinsed and tested in a fresh electrolyte using the same procedure, but without the Cu complex added. Finally, X-ray diffraction (XRD) and scanning electron microscopy (SEM) were employed to characterize the Cu-complex-derived catalysts deposited on the carbon paper electrode.

## 2. Results and Discussion

The redox potential of the five Cu complexes was investigated by cyclic voltammetry using 1 mM Cu complexes dissolved in 0.1 M TBAP and DMF solution with 0.1 M deionized (DI) water in argon. Figure 2 shows the third cycle (see three consecutive cycles in Appendix A) of current density versus potential curves. The four Cu complexes (*dmp, hp, hbpy, htpy*) showed a quasi-reversible redox wave within the −2.4 to 0.52 V vs. Fc/Fc^+^ potential range. Based on the known electrochemical behaviour of these complexes, the redox reaction was assigned to the Cu^2+^ to Cu^+^ redox reaction, modulated by the different coordination of the ligands. The redox potential of the heteroleptic Cu complexes (*hp, hbpy, htpy*) shifted to a more negative potential as compared to that of the homoleptic Cu complex (*dmp*) in Table 1. In simple terms, the negative shift can be rationalized by the stronger coordination of the HETPHEN ligand as compared to the *dmp* ligand. Among the heteroleptic complexes, *hbpy* showed a 70 mV negative shift as compared to *hp*, consistent with the redox potential difference between the homoleptic analogues of *bpy* versus *dmp* complexes [38]. The Cu^2+/1+^ redox potential of *htpy* showed a larger negative shift of 450 mV as compared to the *hp*, which is attributed to the enhanced electron density on the Cu^2+^ due to the coordination of the third N of the *terpy* ligand. The redox potential of the Cu^2+/1+^ of *cq* is even more negative at −1.27 V, as previously reported [37]. The five complexes provide a redox potential range of almost 1.3 V. In the following sections, the effect of this large redox potential variation on CO_2_ reduction performance is investigated.

In Appendix A, the CVs of five Cu complexes recorded under argon and CO_2_ atmospheres in the −2.37 to −0.672 V vs. Fc/Fc^+^ potential range are shown. The CVs measured in CO_2_ show sharply rising current densities with no discernible redox peaks, while the CVs under argon atmosphere show a diverging behaviour with some samples showing a redox peak at highly negative potentials indicative of ligand reduction (*dmp-A* and *dmp-B*) and Cu^2+^ reduction (*cq*) with a possible contribution from the deposition of the Cu-complex-derived catalyst and/or electrolyte decomposition. This potentials range is more negative than the Cu^2+/1+^ redox peaks identified under an argon atmosphere in Appendix A except for the *cq* complex. Within this range, water reduction to H_2_, CO_2_ reduction to various products, as well as ligand reduction and deposition of a Cu-complex-derived catalyst are all possible reactions contributing to the current [28]. Next, chronoamperometric measurements were performed at −2.17 V vs. Fc/Fc^+^ under a CO_2_ atmosphere and the products of CO, CH_4_, and H_2_ were collected using the GC after 20, 40, 60, and 80 min, as shown in Appendix A. The catalytic conversion of CO_2_ to various products could be a superposition of a homogeneous and heterogeneous catalytic mechanism, as explained above, i.e., it may involve the molecular catalyst of Cu^2+/1+^ complexes dissolved in electrolytes or the catalytic activity of a Cu-complex-derived catalyst deposited during the course of the chronoamperometric measurements. It is expected that towards the end of the testing, heterogeneous catalysis may start to dominate, as the coverage of the carbon paper electrode with the catalyst deposited in situ is more compact. The ratio of FE values determined for CO, CH_4_, and H_2_ after the first 20 min of chronoamperometry as well as after 80 min in Table 1 were correlated with the redox potential determined by CV, as shown in Figure 3a–f. The correlation in Figure 3a shows a distorted bell-shaped curve, i.e., as the redox potential decreases, the FE ratio for CO at 20 min increases from 14.6% to as high as 57.7% in the order of *dmp-A*, *dmp-B*, *hp*, and *hbpy* (see line drawn to guide the eye). However, the Faradaic efficiency ratio for CO decreases for *htpy* and *cq* complexes. The distorted bell-shape curve can be explained by a homogeneous CO_2_ to CO conversion mechanism involving the *dmp*, *hp*, and *hbpy* complexes. In these complexes, increasing the electron density on the Cu^2+/1+^ (easier to oxidize) facilitates that conversion of CO_2_ to CO, possibly through enhanced CO_2_ binding to Cu ions. The drop in performance for the *htpy* complex can be explained by the weakened coordination of CO_2_ due to the more crowded Cu^2+^ coordination sphere occupied by the extra N atom at the fifth coordination site of the *terpy* ligand. The homogeneous *cq* complex catalyzes water reduction to H_2_ over CO_2_ reduction to CO, as shown in Figure 3b, which could be attributed to its ability to bind water molecules to the Cu^2+^ site [39]. Note that about 30 times more H_2_ was collected in the presence of *cq* molecular catalyst as compared to that of using carbon paper alone at the same applied potential (Appendix A). Two H_2_O molecules can bind to the Cu site, enhancing water reduction. A distorted bell-shaped curve excluding *cq* complex is therefore clearer (Appendix A), as the other Cu complexes favour catalyzing CO_2_ reduction. In contrast, Figure 3c shows an inverse bell-shaped curve, i.e., as the redox potential increases, the FE for CH_4_ at 20 min increases in the order of *hbpy, hp, dmp-A*, and *dmp-B* complexes. Similarly to CO in Figure 3a, the FE for CH_4_ is lower for the *htpy* and *cq* complexes. In a previous work using the *dmp* complex, we found that the main product of CO_2_ reduction switched from CO to CH_4_ during chronoamperometry. CH_4_ production was attributed to a heterogeneous catalytic mechanism driven by the deposition of a Cu-complex-derived catalyst [30]. The inverse relationship between the redox potential and CH_4_ evolution could therefore be attributed to the rate of forming a heterogeneous catalyst on the working electrode surface as well as its catalytic activity. The more positively shifted the redox potential of the Cu^2+/1+^ is, the less stable the complex is, and therefore, the faster the formation of the Cu-complex-derived catalyst is. We studied the correlations between the redox potential and the specific capacitance, the current density in CV, and the total charge during CA in Appendix A. The specific capacitance (F/g) was calculated at the same potential (0.2 V) in Appendix A of the Cu-complex-derived catalysts using the CV data in Appendix A. We could not calculate the specific capacitance of the sample derived from the *hp* complex due to the broad redox peaks at 0.2 V. We could not see any significant correlations in Appendix A, which implies that other factors, i.e., the solubility of the ligands or the reduced Cu complex, also contribute to the rate of catalyst decomposition. *dmp-B* and *dmp-A* complexes showed the highest FE ratio for CH_4_ as 72.4 and 69.6% at 20 min, while the FE ratio for CH_4_ using the other complexes was less than 52%. This suggests that the HETPHEN strategy is better for CO_2_ reduction to CO in a homogeneous mechanism than depositing a heterogeneous catalyst performing CH_4_ production. In summary, the increased stability of the Cu complex with more negative redox potential may explain the preference of CO production by the Cu complex molecular catalyst, while the decreased stability of the Cu complex with a more positive redox potential may explain the faster deposition of the Cu-complex-derived catalyst leading to the preference for CH_4_ production via a heterogeneous mechanism. The correlation between the FE ratio of gas products at 80 min of CA and the redox potential in Figure 3d–f shows similar trends as compared to that in Figure 3a–c except for *cq* complex. The similar trends in product FE ratios at 20 min and 80 min in Figure 4a–c suggest that even at higher heterogeneous catalyst loading at 80 min, the conversion of CO_2_ to CO and H_2_ evolution from water still proceed through a homogeneous mechanism, while CH_4_ is produced by a heterogeneous mechanism involving the Cu-complex-derived catalyst. However, the changes in the behaviour using the *cq* molecular catalyst, i.e., the increased FE ratio for CO from 17.1% to 29%, indicates a switch to a heterogeneous mechanism for this catalyst. The purely heterogeneous mechanism was tested by performing CA using the working electrodes with the deposited Cu-complex-derived catalyst in a fresh 0.1 M TBAP/DMF electrolyte without the Cu complex. The amount of gas products detected using the GC are shown in Appendix A. XRD and SEM measurements show the morphology of the Cu-complex-derived catalysts in Appendix A, respectively. Figure 3g–i shows the correlation between the redox potential of the Cu^2+/1+^ complex used to deposit the catalyst and the catalytic performance of the Cu-complex-derived catalysts in the purely heterogeneous mechanism. Unlike in Figure 3a,b, no clear correlation between the FE ratio for CO and the redox potential of the original Cu^2+/1+^ complex is found in Figure 3g. This suggests that the mechanism switched from a homogeneous to a heterogeneous mechanism for CO in the absence of a dissolved molecular catalyst. The CO product ratio increased for *cq, dmp-B, htpy*, and *hp*, while it decreased for *dmp-A* and *hbpy*. The main difference between the highly efficient CH_4_ catalyst *dmp-A* sample is the ability to maintain high CH_4_ FE while *dmp-B* partially produced CO after the *dmp* complexes were removed from the electrolyte. The heterogeneous catalysis results suggest that CO_2_ binding strength to the new Cu-complex-derived catalyst is unrelated to the redox potential of the Cu^2+/1+^ ion. This is not surprising since XRD shows the presence of Cu metal, hence the binding energy should be similar provided that the CO_2_ binds to the same crystal facet. The FE ratio for H_2_ decreased in all samples in Figure 3h in the purely heterogeneous mechanism, confirming that H_2_ evolution predominantly followed a homogeneous mechanism, shown in Figure 3b,e. The correlation between the redox potential and FE ratio of CH_4_ in the purely heterogeneous mechanism still exists in Figure 3i but it seems less correlated as compared to Figure 3c,f. The weaker linear correlations in product FE ratio for CH_4_ between the purely heterogeneous mechanism and at 20 min and 80 min of CA in the Cu complex dissolved electrolyte (Appendix A) confirmed that *dmp-B* and *hbpy* are out of the linear correlations. *dmp-B* produced more CO than *dmp-A* in the purely heterogeneous mechanism, leading to the difference in FE ratio for CH_4_ (20%). *dmp-A* and *dmp-B* showed the same redox potential of Cu^2+/1+^ in CV; however, showing different selectivity for CO and CH_4_ indicates that other factors may affect the product selectivity, i.e., by changing the morphologies or crystal facets during homogeneous catalysis. The stability of the *dmp*-complex-derived catalyst was tested for 5 h [28]. The current gradually decayed and the FE for CH_4_ decreased over time. The decaying performance was thought to originate from the restructuring of the catalyst, evidenced by the morphological change before and after the long-term test. *hbpy* in the purely heterogeneous mechanism would be more active to produce CH_4_ compared to *hp* and *htpy*.

The relationship of the redox potential of the Cu^2+/1+^ complexes and their product selectivity for CO_2_ reduction is shown to reveal previously unknown correlations, i.e., a distorted bell-shaped curve. This finding does not follow the correlation found in previous studies between the molecular structure (electron-donating or -withdrawing groups) and the CO_2_ reduction performance using non-Cu-based complexes [8,14,15]. A possible reason for the more complex behaviour is the interplay between the transformation of the Cu complex to the Cu-complex-derived catalyst and CO_2_ reduction to CO, both affected by the redox potential of the Cu^2+/1+^ complexes. This study is among the first attempts to find correlations between the redox potential and CO_2_ reduction performance using Cu^2+/1+^ complexes. We emphasize that from the point of view of designing new complexes with improved performance, not only the correlations between the redox potential and CO production using the Cu complex catalyst in homogeneous mechanisms, but also the lack of correlations with the catalytic performance in purely heterogeneous mechanisms, are also important. The *cq* complex catalyst performed highly efficient hydrogen evolution (FE ratio for 53.1% at 20 min of CA). The homogeneous *cq* complex catalyst could be further explored for hydrogen evolution at the conditions not forming Cu deposits, i.e., lower negative potential or aqueous solution. The correlations in trying to separate homogeneous and heterogeneous mechanisms will be further studied by the new techniques, i.e., in situ and in operando spectroscopic techniques (in situ FT-IR, XPS, XAS) and quantum chemical methods to investigate the catalyst structures and catalytic reaction mechanisms. Those techniques will create big data sets, i.e., crystallographic data, morphology, Cu-Cu coordination number, and Cu-ligand bonding energy. Our approach of using the correlation analysis of five molecular structures should be expanded by using machine learning algorithms on big data sets by studying hundreds of molecular structures as the next step. Such big-data research methodologies using advanced statistical tools and machine learning algorithms are expected to dominate the development of redox-active catalysts in the future. By utilizing those techniques, the origin of catalytic performance of the macrocyclic Cu complexes, which are dominated by ligand reduction and are known to form reversible Cu-complex-derived catalysts during CA, could also be also investigated. Separating homogeneous and heterogeneous mechanisms using correlation analysis will help researchers design Cu complexes specifically tailored for homogeneous or heterogeneous catalysts.

## 3. Materials and Methods

### 3.1. Materials

*N,N*-dimethylformamide (DMF, RCI Labscan, Bangkok, Thailand, 99.8%), tetrabutylammonium perchlorate (TBAP, Merck, Darmstadt, Germany, 98%), were commercially obtained.

### 3.2. Preparation of Cu Complexes

Bis(1,10phenanthroline) iodo copper(II) TFSI (*dmp*) [40], bis(8-quinolinolato) diaqua copper(II) (*cq*) [39], and 6,6′-bismesityl-2,2′-bipyridine [41] were synthesized according to previous studies. Other compounds were commercially obtained and used as received.

#### 3.2.1. (6,6′-Bismesitil-2,2′-bipyridine) (1,10-phenanthroline) Acetonitrile Copper (II) TFSI (*hp*)

Copper (II) perchlorate hexahydrate (182 mg, 0.49 mmol) was dissolved to degassed acetonitrile (10 mL) then solution of 6,6′-bismesityl-2,2′-bipyridine (200 mg, 0.51 mmol) in DCM (10 mL) was added under argon. The resulting mixture was stirred at r.t. for 10 min then 1,10-phenanthroline (92 mg, 0.51 mmol) was added, followed by lithium TFSI (1.564 g, 5.00 mmol). The stirring was continued for 10 min then the solvents were removed under vacuum. The remaining oil was treated with diethyl ether (80 mL) and the solution was kept in the freezer overnight. The green crystals were filtered off, washed several times with diethyl ether and dried.

Yield: 81%; Elemental Analysis: Calc. for C_46_H_39_N_7_F_12_O_8_S_4_Cu [%]: C 44.64, H 3.18, N 7.92. Found: C 45.10, H 3.29, N 8.12. m.p. 200 °C dec. (loss of solvent 110–121 °C).

#### 3.2.2. (6,6′-Bismesitil-2,2′-bipyridine) (2,2′-bispyridine) Acetonitrile Copper (II) BF_4_ (*hbpy*)

The compound was synthesized similarly to *hp* except using Cu(BF_4_)_2_ xH_2_O, 2,2′-bispyridine and skipping anion exchange step.

Yield: 81%; Elemental Analysis: Calc. for C_40_H_38_B_2_N_5_F_8_Cu [%]: C 58.17, H 4.64, N 7.69. Found: C 57.87, H 4.73, N 7.51. m.p. 281 °C dec.

#### 3.2.3. (6,6′-Bismesitil-2,2′-bipyridine) (2,2′:6′,2″-terpyridine) Copper (II) BF_4_ (*htpy*)

The compound was synthesized similarly to *hp* except using Cu(BF_4_)_2_ xH_2_O, 2,2′:6′,2″-terpyridine and skipping anion exchange step.

Yield: 77%; Elemental Analysis: Calc. for C_43_H_38_B_2_N_5_F_8_Cu [%]: C 59.92, H 4.44, N 8.13. Found: C 59.79, H 4.48, N 8.01. m.p. > 300 °C.

### 3.3. Electrochemical Measurements

Electrochemistry was conducted in a three-electrode, two-compartment cell (Pine Instruments, Durham, NC, USA) with a potentiostat (650D, CHInstrument). A carbon paper electrode was employed as the working electrode in a compartment and Ag/AgNO_3_ was used as the reference electrode in the same compartment. A platinum mesh was used as the counter electrode in another compartment. Two compartments were separated by a glass frit. Surface area of carbon paper electrode was 1.5 cm^2^. An amount of 1 mM Cu complexes were dissolved in DMF, 0.1 M TBAP and 0.1 M water. Before the measurement, the cathodic compartment was purged by argon for 30 min. After CV measurement in argon, the cathodic chamber was purged by CO_2_ for 30 min and CO_2_ gas was bubbled with a 15 mL/min constant flow rate during CV and CA measurement. The electrolyte was stirred with 300 rpm during electrochemical measurement. The redox potential of the Cu complex was determined by using oxidation and reduction potentials in Figure 2 using the equation E_redox_ = (E_oxi_ + E_red_)/2. *hp* and *hbpy* complexes showed two oxidation and reduction potentials. A reduction and an oxidation peak at more negative potential were due to the presence of possible homoleptic complexes [34]. The redox potential of *hp* and *hbpy* complexes was therefore calculated by the more positive potentials. All potentials applied in this work were adjusted to the potential versus Fc/Fc^+^ added as an internal standard.

### 3.4. Gas Product Analysis

The gaseous products were measured through the gas outlet of electrochemical cell to a gas chromatograph (GC2030, Shimadzu, Kyoto, Japan) equipped with thermal conductivity (TCD) and flame ionization detectors (FID). The amount of gas products was calculated by calibration curves of known volumes of gas.

### 3.5. Material Characterization

X-ray diffraction (XRD) patterns of rinsed carbon paper electrodes were measured with a PANalytical Empyrean goniometer with a long focus Cu anode tube. The accelerating voltage was 45kV and the current was 40 mA in a standard Brag–Brentano reflection geometry. All scans were performed in a 2theta angle range of 20 to 90°. Field-emission scanning electron microscopy (FE-SEM, JEOL, JSM-7500FA) was used to obtain the morphologies of deposits on carbon paper electrode.

## 4. Conclusions

In summary, we investigated whether controlling the redox potential of Cu^2+/1+^ complexes influences their CO_2_ reduction performance for the first time. We have tested five Cu complexes with different ligands, including new HETPHEN complexes and Cu-(bis(8-quinolinolato), showing a wide range of redox potentials of Cu^2+/1+^. We have found that the redox potential significantly affected the FE ratio of CO production by the Cu complex homogeneously dissolved in electrolyte. Even through a Cu-complex-derived catalyst was formed during CA, homogeneous catalysis of CO_2_ to CO by the Cu complex was still dominant except for the *cq* complex. The *cq* complex catalyst mainly performed the hydrogen evolution by binding water molecules. CH_4_ was produced by the Cu complex-derived catalyst formed in situ, and *dmp-A* and *dmp-B* showed the highest FE ratio for CH_4_. HETPHEN complexes performed lower FE for CH_4_ but higher FE for CO than that of *dmp* complexes, suggesting that HETPHEN complexes are better for performing homogeneous catalysis for CO production. In purely heterogeneous mechanism, there was no correlation between the redox potential and the FE ratio for CH_4_. CH_4_ production is probably controlled by the structure and morphology of the Cu-complex-derived catalyst rather than the redox potential of Cu^2+/1+^. This study opens the possibility of systematic studies between molecular structure, transformation of Cu complexes, and CO_2_ reduction performance, by the benefit of machine learning algorithms, for the design of new homogeneous and heterogeneous Cu complexes.

## Figures and Tables

**Figure 1 molecules-28-05179-f001:**
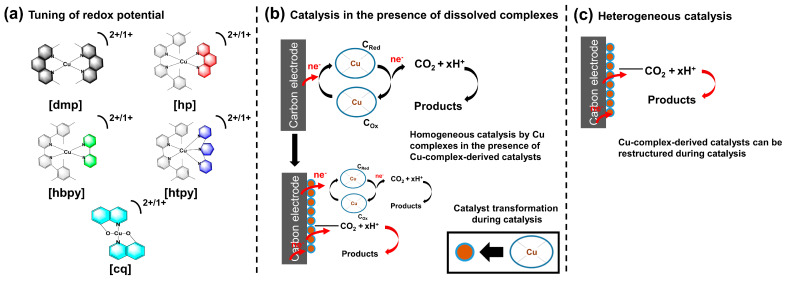
(**a**) The molecular structures of the Cu^2+/1+^ complexes, (**b**) CO_2_ reduction by Cu complexes dissolved in electrolytes, (**c**) CO_2_ reduction by Cu deposits (heterogeneous catalysis).

**Figure 2 molecules-28-05179-f002:**
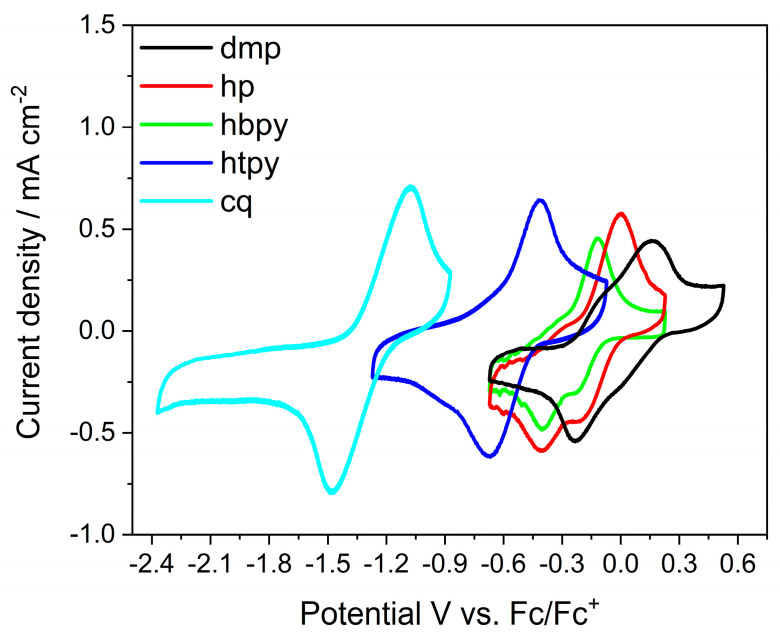
Cyclic voltammetry curves of five Cu complexes representing the Cu^2+/1+^ redox reaction measured in 0.1 M TBAP/DMF with 0.1 M water under argon atmosphere.

**Figure 3 molecules-28-05179-f003:**
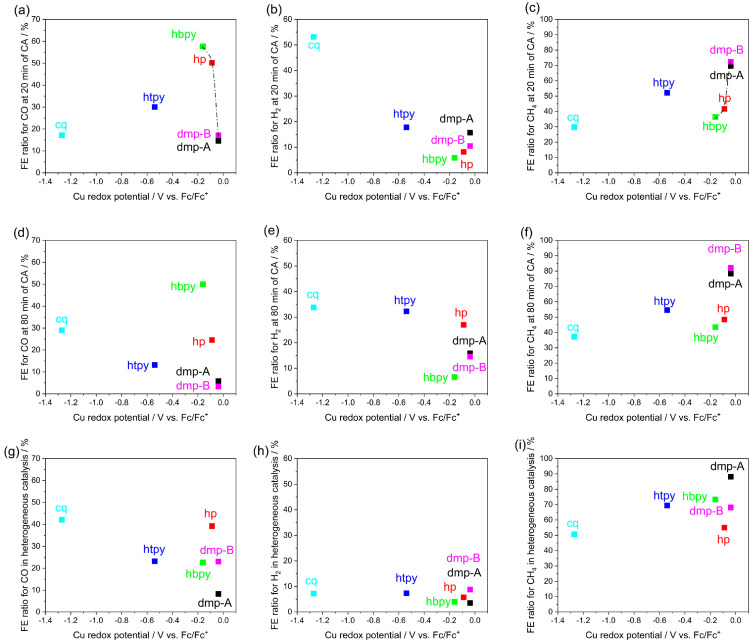
The Cu redox potential in argon versus Faradaic efficiency ratio of reduced gas products (CO, H_2_, CH_4_) at 20 min (**a**–**c**) and 80 min (**d**–**f**) and in heterogeneous catalysis (**g**–**i**) of CA at −2.17 V vs. Fc/Fc^+^.

**Figure 4 molecules-28-05179-f004:**
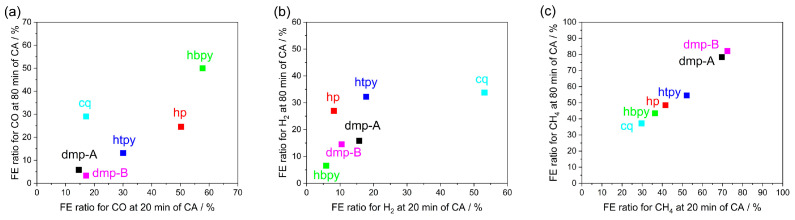
FE ratio at 20 min versus at 80 min of CA for CO (**a**), H_2_ (**b**), and CH_4_ (**c**).

**Table 1 molecules-28-05179-t001:** Redox potentials of Cu complexes in argon and the gas product ratio of Faradaic efficiencies during CA in CO_2_.

Cu Complex	Cu^2+/1+^ Redox Potential (V vs. Fc/Fc^+^) in Argon	Cu^2+/1+^ Redox Potential (V vs. NHE) in Argon	FE Ratio for CO (%) (20 min)	FE Ratio for H_2_ (%) (20 min)	FE Ratio for CH_4_ (%) (20 min)	FE Ratio for CO (%) (80min)	FE Ratio for H_2_ (%) (80min)	FE Ratio for CH_4_ (%) (80min)	FE Ratio for CO (%) (Hetero)	FE Ratio for H_2_ (%) (Hetero)	FE Ratio for CH_4_ (%) (Hetero)
*dmp-A*	−0.039	0.68	14.6	15.7	69.7	5.8	15.8	78.3	8.3	3.5	88.2
*hp*	−0.089	0.63	50.2	8.2	41.6	24.6	27.0	48.5	39.2	5.8	55.0
*hbpy*	−0.161	0.56	57.7	5.9	36.4	50.0	6.6	43.5	22.6	4.0	73.4
*htpy*	−0.54	0.18	30.0	17.8	52.2	13.2	32.3	54.6	23.2	7.3	69.5
*cq*	−1.27	−0.55	17.1	53.1	29.8	29.0	33.8	37.2	42.1	7.2	50.7
*dmp-B*	−0.039	0.68	17.1	10.5	72.4	3.4	14.5	82.1	23.0	8.8	68.2

## Data Availability

Data reported in the manuscript will be available upon reasonable request to the corresponding author.

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
