# Peer review of "Effect of the Cu^2+/1+^ Redox Potential of Non-Macrocyclic Cu Complexes on Electrochemical CO_2_ Reduction"

_molecules, 2023, doi:10.3390/molecules28135179_

Round 1
Reviewer 1 Report
In this work, the authors report a Cu2+/1+-complex using the HETPHEN strategy for CO2 reduction reaction and investigate whether the redox potential of the complexes has any significant effect on CO2 reduction performance. That dmp-B complex exhibits a FE ratio of 82.1% for CH4 at -2.17 V vs. Fc/Fc+ in 0.1 M TBAP/DMF solution. The following comments need to be addressed before the work can be further considered for publication.
1. What is the meaning of “HETPHEN”? The authors should provide the full name of HETPHEN.
2. Please provide the long-term stability of dmp-B complex at -2.17 V vs. Fc/Fc+ in 0.1 M TBAP/DMF solution under CO2 atmosphere.
3. Besides SEM images of carbon paper electrodes with deposits after CA, TEM results need to be provided in the manuscript.
4. Double-layer capacitance is positively associated with the ECSA, which is an important parameter to evaluate the intrinsic activity of electrocatalysts. Moreover, the intrinsic activity data, such as ECSA normalized current density, should be provided.
5. The linear sweep voltammetry curves of five Cu complexes in 0.1 M TBAP/DMF under argon and CO2 atmosphere should be added.
6. To attract a much wider readership, more work about heterogenous electrocatalyst is suggested to be cited: Inorg. Chem. Front. 2023, 10, 2100–2106; Chem. Eng. J. 2023, 454, 140292
Minor editing of English language required.
Reviewer 2 Report
This manuscript develops Cu2/1+-complexes for CO2 electrochemical reduction reaction. Generally, the manuscript merits the readers of the journal of Molecules. Therefore, I would support publishing this paper with minor corrections.
1. The stability of the catalyst should be provided as this factor is the main concern for this study.
2. HETPHEN should be explained at its first use, and this issue should be avioded.
3. The selectivity between CH4 and CO highly relies on the kinetics of the reaciton, I would suggest the author giving further disucssion on this point.
4. The recent publications, such as Sustain. Energ. Fuel, 2020, 4, 5812, and J. Mater. Chem. A, 2020, 8, 4700, merits the general reader of Moleucles.
Reviewer 3 Report
In this work, Kim et al. study the redox potential of various Cu complexes and investigate the existence of possible correlations between such redox potentials and the Faraday efficiency (FE) for the production of methane, CO from CO2 as well as H2 generation. The potential occurrence of correlations between measurable parameters and catalytic performance is surely worth investigating; nevertheless, as found by the authors, there are many additional factors coming in the picture, such as the occurrence of heterogeneous processes, alternative reaction pathways (H2 generation), steric effects of the ligands, adsorption on the electrode, etc. that seriously affect the possibility to establish such correlations. In the end, what one generally finds in this kind of studies is a correlation between a restricted class of structurally similar compounds but the absence of a general correlation. Overall, the work seems well conducted and competently researched so I recommend it for publication in Molecules after major but not extensive revisions as noted below:
a) At Page 3, when introducing the complexes in Figure 1a, the authors should more clearly specify if any of these complexes is new and, if so, report a complete set of characterization data; else, the authors could simply cite a list of previous references for each complex.
b) At Page 8, the authors claim that "The relationship of the redox potential of the Cu2+/1+ complexes and their product selectivity for CO2 reduction is shown to reveal previously unknown correlations, greatly improving our understanding of the homogeneous and heterogeneous CO2 reduction mechanisms using the Cu complexes." The authors do not specify how our knowledge of the mechanisms is greatly improved and this kind of conclusion seems quite hurriedly. To be honest, In the graphics in Figure 3, I do not really see a general correlation, they look more like randomly scattered points inside a square. There might be correlations when selecting specific groups of complexes but, then, the authors should report separate graphics that contain only a selection of the complexes, explain the trend in detail and mention why some specific complexes are not included for not having the specific characteristics required to match the trends. With the graphics as they are reported now, no conclusions can be drawn.
c) As this is not a really new topic and there is a huge amount of available literature, the authors should explain in the conclusion or in the discussion before the conclusions, whether their findings conform with previous literature or not.
The English is very good as written by native speakers. However, even native speakers seem to make mistakes, such as at Page 4, lines 138-139 "tridente" for "tridentate" and "cite" for "site".
Also at lines 217-218 the sentence "cq complex is a homogeneous mechanism catalyzes water reduction..." is kind of awkward.
Round 2
Reviewer 1 Report
The manuscript has been largely completed.
Extensive editing of English language required.
Reviewer 3 Report
Whereas my initial impression that the authors could not establish a strong correlation between the redox potential of the Cu2+/1+ complexes and their product selectivity for CO2 reduction remains, the work is well carried out and the authors competently answered all the reviewers remarks. Therefore, I recommend the publication of this article as is.